# The “Ifs” and “Hows” of the Role of Music on the Implementation of Emotional Regulation Strategies

**DOI:** 10.3390/bs12060199

**Published:** 2022-06-20

**Authors:** Mariana Carvalho, Nicoletta Cera, Susana Silva

**Affiliations:** 1Center for Psychology at University of Porto (CPUP), Faculty of Psychology and Education Sciences, University of Porto, 4200-135 Porto, Portugal; up201604356@edu.fpce.up.pt (M.C.); cera.nicoletta@fpce.up.pt (N.C.); 2Coimbra Institute for Biomedical Imaging and Translational Research (CIBIT), 3000-548 Coimbra, Portugal

**Keywords:** emotional regulation, music listening, musical Sophistication, executive functions, empathy

## Abstract

Music is believed to aid the implementation of emotion regulation strategies like distraction or reappraisal, but empirical studies have shown null results. However, the moderating role of one’s relation with music (musical sophistication) and/or executive functioning skills has not been considered yet. In addition, little is known about how music acts. In the present study, we induced anger in a group of participants characterized by musical sophistication and executive functioning. We asked them to regulate their emotional state and measured regulation efficacy. Participants were split into four groups, defined by regulation strategy (distraction vs. reappraisal) and music (with vs. without). Results indicated music effects in higher, but not in lower musical sophistication participants. In the former, music benefitted from reappraisal but was an impaired distraction. Two different executive functions—working memory and affective flexibility—had opposite effects: higher, but not lower working memory participants benefited from music; lower, but not higher, effective flexibility participants took advantage of music. Reports of subjective experience suggested that music favors more empathic reappraisals, and that these may be more long-lasting. Our findings support the idea that music effects depend on listeners’ characteristics, and they raise new hypotheses concerning the specificity of emotional regulation aided by music.

## 1. Introduction

From time to time, we can feel overwhelmed by our own emotions, and sometimes we only feel relief when we find some control over them. This control can be achieved through the regulation of emotional experience [1], such that our current emotional system or its expected state becomes able to match our emotional goal. Distraction and reappraisal are two strategies of *emotional regulation* (ER). Distraction can be conceived as finding something that takes our mind off the problem situation, whereas reappraisal allows us to see the situation with a fresh pair of eyes. Distraction consists of disengaging attention from the salient aspects of an emotional-eliciting event [2] by focusing on another concurrent task [3]. Conversely, reappraisal implies directing attention to the emotionally eliciting event. Since the emotional impact of an event is mediated by the way in which we interpret or appraise it, the aim of reappraisal is to reinterpret and change the meaning of that event [4].

The efficacy of ER does not exclusively depend on our will. ER through reappraisal needs resources such as time and cognitive capacity [5]. Similarly, distraction involves multiple focuses on environmental stimuli [6]. The behavior we use to distract ourselves or reappraise a given situation may, thus, be crucial for the effective implementation of these strategies.

Music listening is one behavior that is recognized as a powerful tool or tactic for regulating emotions [7,8,9]. Music is known to increase well-being and impact the intensity and expression of emotions [10]. Music listening is used deliberately for several reasons, including the attempt to control, support, or change the current state of experienced emotions [11]. According to the GSTM (goals, strategies, tactics, and mechanisms) framework [8], music listening is frequently referred to as a tactic that supports the implementation of distraction and reappraisal [7,9], and it seems to be particularly associated with reappraisal. For instance, the message and meaning transmitted by the lyrics of a song positively affect the problem solving and the appraisal of a difficult situation [12]. Nevertheless, potentially destructive uses of music, like ruminating [9] and immersing in negative emotions [12] have also been documented. Thus, music listening seems to be an appropriate tool for ER. However, several questions regarding the relation between music listening and ER are still open.

First, it remains unclear whether music listening is a useful tactic for implementing ER strategies. Associations between music listening and ER have been reported in empirical studies (e.g., [12,13]), but most are based on retrospective accounts. Moving away from this retrospective logic, the study of Baltazar et al. [14] focused on the efficacy of music in stress reduction when combined with an ER strategy, considered relevant by participants, in the context of real-time ER. This study showed that combining music listening with an adequate strategy did not increase ER efficacy, when compared with music or strategy alone. However, the fact that participants were asked to choose the more adequate ER strategy elicited a variety of strategies, including potentially negative ones such as rumination (i.e., thinking repetitively and passively about a particular emotional event [15,16]). In addition, the authors did not consider two important individual variables with a potential moderating role in the relation between music listening and ER. The first one is *musical sophistication* [17], conceived as the significance of music in participants’ lives. The second is the level of *executive functioning* of participants. Executive functions are a set of abilities that allow decision making, controlling impulsive behaviors, and cognitive flexibility [18]. The potential moderating role of musical sophistication in music listening-based ER seems straightforward. If music is irrelevant in one’s life, it is unlikely that it will boost ER. To our knowledge, this hypothesis remains untested. Concerning the possible impact of executive functioning (EF) abilities, it has been hypothesized that EF abilities improve ER skills [19,20]. Viewing music listening as a tool that aids ER [7,9] suggests that music listening may compensate for weak EF skills. Therefore, music listening should be more useful to individuals with lower than for those with higher EF skills. Again, to our knowledge, little is known about this possibility. 

Moreover, the chance that music helps the implementation of ER strategies like reappraisal raises questions about the underlying process. One possibility is that music improves empathy. Listening to music has been associated with helping behavior [21]. Specifically, there seems to be an association between listening to music with prosocial lyrics and an improvement in helping behavior. This suggests that this kind of music can influence our cognition and affect, and it is in line with the media effects theory [22]: according to this theory, listeners’ thoughts and feelings are congruent with the message conveyed by music. Since prosocial behavior implies some degree of empathy [23], it can be hypothesized that the key role of music in reappraisal is related to an improvement of empathy. To our knowledge, this hypothesis has not yet been tested.

In the present study, our first goal was to determine whether the effect of music listening on ER is moderated by musical sophistication and/or executive functioning. As a second goal, we wanted to better understand the role played by music in the chosen reappraisal mode—specifically if music promotes empathy-based forms of reappraisal.

To address these goals, we conducted a mixed 2 × 2 × 2 factorial design study with ER type as a within-subjects factor (passive and active), and ER strategy (distraction and reappraisal) as well as music listening (music and no music) as between-subjects. Participants went through a passive ER attempt (PER, no regulation instructions) and an active ER attempt (AER, instructions to regulate). The two ER attempts followed an anger induction task based on autobiographical memories. Before and after anger induction based on an autobiographical memories task, participants self-rated their anger level. In the active condition, participants were randomly assigned to four experimental conditions resulting from the combination of regulation strategy (distraction vs. reappraisal) and music (with vs. without music): distraction with music, distraction without music, reappraisal with music, reappraisal without music. Success in regulation—passive and active—was measured based both on anger ratings and self-reports of perceived success in regulation. To address the first goal, we focused on success in ER during the session (active minus passive regulation) as a function of music listening and regulation strategy to see whether music listening alone influenced ER. Afterwards, we added musical sophistication and executive functioning as moderators, using two new models. In line with the literature and our own analysis of factor structure (see below, methods), executive functioning was approached as a non-unitary construct, where executive functioning tasks do not always show associations among them- mostly in adults [24,25,26]. To address the second goal, we asked participants with vs. without music to report how they reappraised the situation.

Regarding the first goal, we hypothesized that musical sophistication and executive functioning levels would have a moderating effect on ER, meaning that the positive impact of music listening on ER would increase with musical sophistication, and that participants with lower executive functioning skills would be more sensitive to the facilitating effects of music. Concerning the second goal, our hypothesis was that music listeners would implement a form of reappraisal different from the one used by non-listeners. Specifically, music listeners would rely more on empathy.

## 2. Materials and Methods

### 2.1. Participants

The required sample size for this study was estimated with an a priori power analysis (with G*Power 3.1 [27]). To obtain a statistical power of 80% with a medium effect size (*f* = 0.25) and α = 0.05 using a design with four groups and two repeated measurements (active vs. passive regulation), we needed to collect data from at least 36 participants.

Forty-eight healthy participants (10 male) aged between 19 and 33 (*M* = 23.17, *SD* = 3.24) were recruited for this study. All the participants had normal hearing and normal or corrected-to-normal vision. Most of them (93.8%) were college students. Participants were randomly assigned to one of the four experimental groups (Distraction With music, D-W; Distraction Without music, D-Wo; Reappraisal With music, R-W; and Reappraisal Without music, R-Wo, *n* = 12 per group). Two participants from the D-Wo group had to be reallocated at D-W because they used music to distract themselves (Appendix A).

All the participants gave their informed consent, and experimental procedures were approved by the local Ethics Committee (Ref. 2010/09-7).

### 2.2. Instruments

#### 2.2.1. Emotional Regulation during the Session

To characterize the outcomes of emotional regulation during the session, we used the following instruments:

*Self-report measure of state-anger (STAXI-2)*: The state-anger subscale of the State-Trait Anger Expression Inventory-2 (STAXI-2, [28,29]) was used as a report of the level of anger experienced by participants at different moments in the session (see Procedure). STAXI-2 comprises fifteen statements, rated on a 4-point scale ranging from 1 (*nothing*) to 4 (*extremely*). The final score of state-anger is obtained by adding the classification assigned to each statement. Higher scores reveal higher levels of anger.

*Emotional regulation follow-up questions***:** After the attempt of active regulation, participants completed a questionnaire addressing their perceptions about the emotional regulation task and the implemented strategies, providing a quantification of the subjective perceptions of the regulation process (Table 1).

#### 2.2.2. Individual Differences in Relevant Dimensions

*Sociodemographic questionnaire*: it assessed basic sociodemographic information such as age, gender, and schooling.

*Regulating Emotion Systems in Everyday Life inventory* (RESS-EMA [30]). The RESS-EMA is a self-report questionnaire designed to assess the individual propensity to use six different ER strategies (distraction, rumination, reappraisal, suppression, engagement, and relaxation). Participants must rate the extent to which each of the 12 statements accurately describes what they usually do in response to their emotions. Responses are measured on a 5-point scale (1 = *not at all*; 5 = *very much*). The score for each strategy ranges from 2 to 10, with each strategy indexed by two items.

*Difficulties in Emotion Regulation Scale* (DERS [18], adapted by [31]). DERS is a 36-item self-report questionnaire consisting of a global score of emotional regulation difficulties and six subscales that assess specific emotional regulation problems: (a) nonacceptance of emotional responses (6 items), (b) difficulty engaging in goal-directed behavior when distressed (5 items), (c) impulse control difficulties when distressed (6 items), (d) lack of awareness of emotions (6 items), (e) limited access to strategies for regulation (8 items), (f) lack of emotional clarity (5 items). Participants are asked how often each item (e.g., *I am clear about my feelings*) applies to them. The response is given using a 5-point scale (1 = *almost never,* 5 = *almost always*). In this scale, it is possible to obtain a score for each subscale by adding the associated items or a total score, with all 36 items. High scores indicate high emotional regulation difficulties.

*Goldsmiths Musical Sophistication Index* (Gold-MSI- [17], adapted by [32]). Gold-MSI is a self-report inventory that describes the different facets of skilled musical behavior in the general population. The facets of musical behavior are grouped into five dimensions: (1) active engagement, e.g., *Music is kind of an addiction for me—I couldn’t live without it*; (2) perceptual abilities, e.g., *I can tell when people sing or play out of tune*; (3) musical training, e.g., *I can play ___ musical instruments*; (4) singing abilities, e.g., *I can sing or play music from memory*; (5) emotions, e.g., *Music can evoke my memories of past people and places.* Besides these five dimensions, the Gold-MSI also provides a General Musical Sophistication factor that incorporates some items of the above-mentioned scales. Gold-MSI has two parts: In the first part, participants rate their agreement with every statement (31 items) on a 7-point scale, anchored by 1 = *Completely Disagree* and 7 = *Completely Agree*; in the second (7 items) they indicate at which frequency they were involved in music-related activities, e.g., *I have had formal training in music theory for 0/0.5/1/2/3/4–6/7 or more years*. High scores indicate high musical sophistication.

#### 2.2.3. Executive Functioning Tasks

*Inhibitory control (IC):* Inhibitory control was assessed with the Emotional Stroop [33]. Here, participants are exposed to three 100-word sets (positive vs. negative vs. neutral valence) and asked to name the ink color of each word. Since cognitive conflict can arise from semantic-related emotional interference coming from positive or negative words, inhibitory control is required for successful task completion. The magnitude of the Emotional Stroop effect, indicating inhibitory control deficits, is measured by the difference between the number of ink colors named on emotional words and the number of ink colors named on neutral words.

*Working Memory (WM)*: The backward portion of the digit span (WAIS-IV; [34]) was used to assess WM. Each item of this test has two sequences of numbers. Participants are asked to repeat each sequence in reverse order after it is read. To obtain 1 point, participants need to be successful in both spans. Only the backward span was applied because it is the one that is particularly dependent on working memory. WM ability is measured by the number of correct items.

*Affective Flexibility (AF):* Affective flexibility was measured using a design similar to that described by [35]. In this task, participants are asked to classify a series of emotional pictures based on two rules that switch along the task: neutral and affective rules. The neutral rule asks for classifications of the image based on the number of human beings shown, whereas, for the affective rules, the classification is based on valence. Rule switching is unpredictable, and the ability to switch with little cost indicates effective flexibility. The outcomes of this task correspond to switch costs, which are obtained by comparing the reaction times (RTs) on switch trials with the RTs on repetition trials. Besides global switch costs, we calculated specific switch costs. These included *negative non affective switch costs,* which concern switching to the non-affective rule when the picture was negative (NA.N); *positive non affective switch costs* concern switching to a non-affective rule when the picture was positive (NA.P); *negative affective switch costs* concern switching to an affective rule when the picture was negative (A.N); *positive affective switch costs* switching to an affective rule when the picture was positive (A.P). According to [32], effective reappraisal is better predicted by increased AF in switching towards the non-affective rule in the presence of a negative picture (NA.N).

We used a selection of 40 images from the Nencki Affective Picture System (NAPS; [36]). The final set was split into four groups: negative pictures with one or fewer human beings, with a mean valence of 1.75; negative pictures with two or more human beings (mean valence = 1.94); positive pictures with one or fewer human beings (mean valence = 8.16); positive pictures with two or more human beings (mean valence = 7.76). Pictures were presented one at a time, randomly, with free response time and no interval between images. Two signals on the left and right sides indicated the rule to be applied (“+” and “−” to apply the affective rule; “≤1” and “≥2” to apply the neutral rule).

### 2.3. Procedure

Due to the pandemic context, the experiment was conducted via video call using Zoom (version 5.4.6.) with one participant at a time (Figure 1).

After reading and signing the informed consent, participants answered the sociodemographic questionnaire, filled in RESS-EMA and DERS, and performed the three executive function tasks measuring IC, WM, and AF, in counterbalanced orders. To record the accuracy and RTs at the AF task, we used the OpenSesame software (version 3.3.5.), with a link sent to participants through Mindprobe/Jatos (version 3.5.1).

The remainder of the session was structured upon two moments of anger induction: the first followed by passive regulation, common to all participants, and the second by active regulation, differing across groups. For anger induction, an autobiographical memory task was used, where participants received instructions to recall an autobiographical event during which they felt “really angry”. This memory should still bring out those feelings of anger. Participants had three minutes to evoke the episode associated with feelings of anger and then were asked to type a narrative about the evoked memory. The autobiographical memory task was chosen because of its ecological validity and its capacity of inducing angry mood states with changes in valence and arousal [37]. Participants were asked to evoke the same episode in the two anger induction moments. During the second, they were encouraged to elaborate on the previous memory by adding details to the narrative that had been written in the first anger induction moment. We were aware that this elaboration could lead to the decrease of anger in a second anger induction. Nevertheless, we chose this method to control for potential differences between two evoked moments and to counteract the possibility that the participant did not have more than one episode of anger to remember. Between the two core moments of anger induction plus the regulation, participants filled in the Gold-MSI to help them return to baseline. 

After the first anger induction moment, *passive emotional regulation* took place, lasting 5 min. Here, all participants were encouraged to avoid regulation attempts. To counteract the natural tendency for self-regulation, we told them: ‘‘Allow yourself to continue to feel the anger that you were feeling previously about the evoked memory, as you naturally would’’ (see [38]). 

In the *active regulation* task, instructions varied according to the between-subjects condition. Two groups (D-W and D-Wo) were instructed to use distraction (D) strategies and encouraged to deflect their attention to something other than the anger-inducing event (see Appendix D for results). In the D-Wo group, they had to choose an activity unrelated to music, while in the D-W group they were instructed to use a self-chosen song for distraction. The other two groups were instructed to use reappraisal (R), which is typically done by reinterpreting the meaning of an emotional situation [39]. Therefore, we encouraged participants to reinterpret the evoked situation in order to decrease their emotional impact. One group was told to listen to a self-chosen song to help them reinterpret the situation (R-W), while the other was only told to apply the reappraisal strategy (R-Wo, see Appendix A). All participants chose songs in a strict sense—i.e., none used instrumental music. As in passive regulation, all participants were given 5 min to accomplish the task. Some authors claim that ER should be implemented at a specific point of the emotional process to be effective (e.g., [40]), while others do not. For instance, [41] defend that ER can be conceived as a continuous and accumulating process, where ER strategies would be effective at any point from the moment that the emotion-triggering event occurs. Since we could not determine the exact stage of the emotional process in our participants, we follow the suggestion of [41].

Along the session, STAXI 2 was filled in six times (Figure 1): (1) as a first baseline; (2) after the first anger induction, indicating pre-passive-regulation anger; (3) after passive regulation (post-passive-regulation anger); (4) as a second baseline, after Gold-MSI; (5) after the second anger induction, indicating pre-active-regulation anger; (6) after active regulation (post-active-regulation anger). The time gap between (3) and (4) corresponded to the time participants took to fill in the Gold-MSI questionnaire, which was about 5 min. Lastly, participants answered the emotional regulation follow-up questions (see Table 1)**.** The entire experiment took about 50 min to complete. Participants saw the experimenter on screen only when instructions were being provided. During task performance, the cameras from the two parts were shut off. The experimenter was not able to see what participants wrote for the autobiographical memory until the end of the experiment.

### 2.4. Data Analysis

#### 2.4.1. Dependent Variables

The main dependent variable in our study was ER intensity during the session, referring to changes in anger after regulation as measured by STAXI-2. Specifically, STAXI-2 ratings obtained after each anger induction task were subtracted from those obtained before (pre-regulation minus post-regulation, resulting in positive values if anger decreases). Since we had two regulation moments, we had two indices of emotional regulation intensity: PER (passive emotion regulation), expected to be null or negative, and AER (active emotion regulation), expected to be positive (Figure 1). We used ER type (AER vs. PER) as a within-subjects factor to obtain a baseline-corrected measure of ER intensity (how larger AER was compared to PER). We refer to this baseline-corrected measure as *emotional regulation efficacy*.

The second dependent variable was perceived success in emotional regulation. It was obtained from one of the emotional regulation follow-up questions. Specifically, we analyze the answers, yes or no, to the question “Did you feel difficulties in the (active) emotional regulation task?” (Question 1, Table 1). This variable was confronted with changes in STAXI-2 ratings for comparisons between objective efficacy and perceived efficacy.

The last dependent variable concerned our second goal, and it corresponded to the type of reappraisal made by participants. It was accessed by the question “How did you reappraise the emotional situation?” (Question 2, Table 1). Responses to the remaining questions regarding subjective perceptions of the regulation process (questions 3 to 6, Table 1) were aggregated to the analyses of reappraisal modes, as included in our second goal.

#### 2.4.2. Control Analyses

To see if the four groups were equivalent in terms of emotional regulation difficulties and preferred strategies, we computed participants’ scores for DERS and RESS EMA. In this respect, we conducted one-way ANOVAs to control for the presence of between-group statistically significant differences.

To make sure the anger induction worked, we calculated the anger induction intensity based on the STAXI-2 score results and compared these values against zero with one-sample tests. Anger induction measures were obtained by subtracting STAXI-2 ratings at the baselines from ratings after the respective anger induction tasks (pre-passive regulation minus baseline 1 and pre-active regulation minus baseline 2). To make sure we had significant ER intensity in the active regulation task and non-significant ER intensity in passive regulation, we also performed one-sample t-tests against zero for PER and AER. These tests were preceded by a repeated measures ANOVA with STAXI-2 scores in all time points, aiming to test for state-anger variation across the six time points.

For the AF task, we eliminated outlier trials with values above and below a window defined according to the outlier labeling rule: we calculated the upper and lower quartiles for each participant and multiplied these two values by g = 2.20 [42] to establish the demarcation points of the window. We then conducted a paired-sample t-test comparing RTs on switch trials with RTs on repetition trials to check for the presence of significant switch costs in global and specific AF tasks. 

To check whether there was a significant Stroop effect in the emotional Stroop task, we performed a paired-samples t-test comparing the number of ink colors named in neutral vs. emotional conditions (neutral vs. negative; neutral vs. positive).

#### 2.4.3. Music Effects on ER Moderated by Musical Sophistication and Executive Functioning

To test music effects on the implementation of ER strategies, we fit a linear mixed effects regression with ER intensity (an objective measure of regulation) as the outcome variable. ER type (passive, PER vs. active, AER), music (yes vs. no), and strategy (distraction vs. regulation) were entered as fixed factors with all main effects and interactions considered (ER type*music*strategy). Participants were considered random factors. We used the lmer function from the lme4 package [43] from R (version 1.4.1106). To assess the subjective experience of regulation, we analyzed the answers (yes or no) given to the question “*Did you feel difficulties in the (active) emotional regulation task?”*. We performed two chi-squared tests for independence to see if the answers to that question differed according to music (D-W vs. D-Wo and R-W vs. R-WO).

To investigate the moderating role of musical sophistication, we computed another model adding the Gold-MSI total score as a fourth fixed factor (ER type*music*strategy*Gold-MSI). The two models were compared using the ANOVA function. Comparisons were made based on AIC (Akaike Information Criterion) values [44]. Significant interactions engaging ER type were broken down into further analyses by subsets. Critical alpha levels were set to 0.05.

Similarly, we tested a model adding executive functioning to the previously described 2 × 2 × 2 basic model. To verify the expected non-association between executive functioning tasks, we computed the correlations among the three tasks (inhibitory control/emotional Stroop, digit span, and affective flexibility in its four variants). None reached significance (*p* > 0.07 before correction for multiple comparisons, see Appendix A). Executive Functioning Tasks). We also ran an exploratory factor analysis which, in line with the correlation results, showed no aggregation of any of the six tasks into the same factor (see also Appendix A). Therefore, we approached executive functioning measures as separate variables. To reach the best model, we made multiple model comparisons engaging up to two executive-functioning-related covariables. (Appendix A) and adopted the model with the lowest AIC [44]. The linear mixed effects model incorporating executive functioning had ER type, music, strategy, and executive functioning variables as fixed factors, and participants as random intercepts. Critical alpha levels were set to 0.05.

#### 2.4.4. Reappraisal Mode with vs. without Music

Focusing on the active ER task using reappraisal, we analyzed the answers to the question “*How did you reappraise the emotional situation?*” (Question 2, Table 1) to find out if participants in the R-W condition reappraised the emotional situation in a different way when compared to participants in the R-Wo condition. With this intention, we made a content analysis, and we counted the references that were inserted in each category (R-W vs. R-Wo). The content analysis followed a deductive procedure, based on the a priori category of interest empathy, mixed with an inductive logic that allowed new categories to emerge from the remaining answers. Alongside, we analyzed the features that participants had focused on during the reappraisal task (with and without music; question 6, Table 1).

Lastly, and following the idea that the way participants reappraise a situation would change depending on the presence of music, we examined if reappraisal modes associated with music listening would favor an ER that extends over time, in other words, we analyzed the answers given to the question (3, Table 1) *“In the future, if you remember that emotional situation again, the levels of anger evoked by her would be the same?*” and (5, Table 1) *“Do you think you will be able to deal better with the emotional situation?”*.

## 3. Results

### 3.1. Control Analyses

Comparisons across the four groups were carried out for emotional regulation abilities and strategies (Table 2). Results showed no significant differences.

Concerning STAXI-2 values across the experiment, the repeated measures ANOVA showed statistically significant differences across the six time points, *F* (5235) = 43.885, *p* < 0.001, thus justifying further local comparisons. Regarding anger induction (Table 3), one sample *t*-tests showed that it was significantly above zero in anger induction 1 (*M* = 0.699, *SD* = 0.649), *t*(47) = 7.45, *p* < 0.001, *d* = 1.08, and anger induction 2, (*M* = 0.628, *SD* = 0.679), *t* (47) = 6.40, *p* < 0.001, *d* = 0.923. As for the effect of passive vs. active ER tasks, the results from one sample *t* tests showed that AER was significantly above zero, *t*(47) = 6.80, *p* < 0.001, *d* = 0.981, meaning that there was a change between pre- and post-active ER moments (*M* = 0.580, *SD* = 0.591). PER was not significantly different from zero, *t*(47) = −0.89, *p* = 0.374, *d* = −0.130, meaning that there was a non-significant negative change (*M* = −0.061, *SD* = 0.471) between pre- and post-passive ER moments. 

Regarding the presence of Stroop effect, results of a paired sample t-test showed that the number of ink colors in neutral words read by participants (*M* = 59.3, *SD* = 16.1) in 45 s was significantly higher than the number of ink colors of negative words (*M* = 52.2, *SD* = 15.3), *t*(47) = 6.64, *p* < 0.001, *d* = 0.945, indicating the presence of a significant emotional interference of negative words, and thus a significant Stroop effect. We did not find a significant difference for the positive words (*M* = 60.8, *SD* = 15.7) compared to neutral words (*M* = 59.3; *SD* = 16.1), *t*(47) = -1.39, *p* = 0.171, *d* = 0.211.

Concerning switch costs and as a result from the paired samples t-test, we found that the global RTs on switch trials (*M* = 2483, *SD* = 1061) were significantly longer than the global RTs on repetition trials (*M* = 2090, *SD* = 7163), t (935) = −18.4, *p* < 0.001, *d* = 0.077, confirming the presence of switch costs.

### 3.2. Music Effects on ER

Concerning the subjective experience of music effects on ER, Chi-squared tests indicated that, in the distraction condition (*n* = 24), responses to the question “Did you feel difficulties in the (active) emotional regulation task?” did not differ as a function of listening to music, χ2(1, 24) = 0.098, *p* = 0.754. Conversely, in the reappraisal condition (*n* = 24), participants who listened to music showed a significantly larger number of “no difficulties” responses, χ2(1, 24) = 6.171, *p* = 0.013.

Regarding music effects as measured by STAXI-2 changes, the basic model (ER type*music*strategy) showed a significant main effect of ER type (*Beta* = − 0.68; *CI* = −1.11–−0.25; *p* = 0.002, Table 4), showing a generalized success in ER (AER more successful than PER). There were no significant interactions, indicating no evidence in favor of music or strategy effects on ER intensity.

### 3.3. Music Effects on ER Moderated by Musical Sophistication

The model adding Gold-MSI as a fourth factor (ER type*music*strategy*Gold-MSI) showed improved fit compared to the basic model (χ^2^(8) = 22.5, *p* = 0.004, AIC = 161.02 vs. 167.53). Along with a main effect of ER type (*Beta* = 2.37; *CI* = 1.13–3.60; *p* < 0.001, Appendix B, Table A1), we found several significant interactions, including the highest-level interaction ER type x music x strategy x Gold-MSI (*Beta* = −0.05; *CI* = −0.09–−0.01; *p* = 0.009). We broke down the latter by analyzing high- vs. low-Gold-MSI participants, divided according to a median split. 

Low-Gold-MSI (Figure 2) showed a significant interaction for ER type x strategy (Appendix B, Table A2). Further analyses considering distraction and reappraisal separately showed significant effects of ER type (significant ER efficacy) in *reappraisal* (*Beta* = −1.16; *CI* = −1.84–−0.48; *p* = 0.001), but not in *distraction* (*Beta* = −0.32; *CI* = −0.78–0.15; *p* = 0.178). None of the two analyses showed significant interactions between ER type and music. In sum, Low-Gold-MSI participants were able to achieve an effective regulation only by using reappraisal, but music was irrelevant for success.

High-Gold-MSI participants (Figure 2) showed a significant main effect of ER type. All the interactions were significant (Appendix B, Table A3), indicating a relevant role for music. We broke down the highest-level ER type x music x strategy interaction by conducting four analyses, one per music x strategy condition (D-W, D-Wo, R-W, R-Wo). All four analyses showed significant effects of ER type (D-W: *Beta* = −0.51; *CI* = −0.90–−0.13; *p* = 0.008; D-Wo: *Beta* = −1.51; *CI* = −2.87–−0.16; *p* = 0.028; R-W: Beta = −0.68; *CI* = −1.12–−0.24; *p* = 0.002; R-Wo: *Beta* = −0.50; *CI* = −0.82–−0.17; *p* = 0.002). However, looking at effect sizes as indicated by estimates, we saw different patterns for distraction vs. reappraisal: in *distraction*, participants showed an increased effect size (*Beta* = −1.51) *without* music (vs. with music, *Beta* = −0.51), while in *reappraisal* the effect size was larger *with* music (*Beta* = −0.68, vs. without *Beta* = −0.50).

In summary, while music was irrelevant to low-Gold-MSI participants, it had some influence on those with higher musical sophistication levels. In the latter, music benefitted reappraisal but harmed distraction.

### 3.4. Music Effects on ER Moderated by Executive Functioning

The best model among all comparisons engaging up to two EF variables was the one with working memory as measured by digit span (DS) and NA.N (negative non-affective switch costs) as covariates (Appendix A). Low NA.N means lower switch costs, hence high effective flexibility. The analysis showed a significant ER type*music*working memory interaction (*Beta* = −0.67, *CI* = −1.19–−0.16, *p* = 0.011, Appendix C, Table A4). Therefore, we broke down the analysis into low- vs. high-working-memory participants (ER type*music*strategy*affective flexibility), based on a median split. Due to the characteristics of the scores (integers, giving rise to several instances of median = 4), two unequal groups were generated (low DS, *n* = 44; high DS, *n* = 52).

*Low DS* participants (see Figure 3) showed a significant interaction between ER type, music, and strategy (*Beta* = −1.25, *CI* = −2.46–−0.03, *p* = 0.044, Appendix C, Table A5). We, therefore, broke down the analysis further, considering distraction vs. reappraisal. For *Low DS/distraction*, ER type interacted significantly with music (*Beta* = 1.11, *IC* = 0.20—2.03, *p* = 0.017). Breaking down the analysis further, we observed significant ER intensity for both Low DS/distraction with music (*Beta* = −0.47, *CI* = −0.85–−0.10, *p* = 0.013).) and Low DS/distraction without music (*Beta* = −1.59, *CI* = −2.66–−0.51, *p* = 0.004).), but the estimate was larger for distraction without music. In contrast, *Low DS/reappraisal* participants showed a significant effect of ER type (*Beta* = −0.74, *CI* = −1.13–−0.36, *p* = < 0.001) without further interactions. Thus, for participants with low DS (low working memory resources), music seems to be relevant when they try to distract, but not when reappraising, and distraction seems to decrease with music.

*High DS* participants showed a significant ER type*music*affective flexibility interaction (*Beta* = −0.00, *IC* = −0.00–−0.00, *p* = 0.012, Appendix C, Table A6). We split the sample into high- vs. low-affective-flexibility individuals, calculating a model for each group. 

*High DS/high-affective flexibility* (Figure 3) participants showed a significant effect of ER type (*Beta* = −0.77, *CI* = −1.50–−0.03, *p* = 0.04) without further interactions. In contrast, the *High DS/low-affective-flexibility* (Figure 3) showed an interaction between ER type, music, and strategy (*Beta* = 0.78, *CI* = 0.15–1.42, *p* = 0.016). We broke down the analysis further by strategies and found that the *High DS/low-affective-flexibility/distraction* group was affected by a significant ER type*music interaction (*Beta* = −0.95, *CI* = −1.44–−0.45, *p* = < 0.001). Splitting the analysis by music, we found significant ER type effects in both conditions (With music: *Beta* = −1.10, *CI* = −1.68–−0.52, *p* = < 0.001; Without music: *Beta* = −0.15, *CI* = −0.26–−0.04, *p* = 0.006), but an increased effect size when music was present. As for *High DS/low-affective-flexibility/reappraisal,* the effect of ER type was non-significant (*Beta* = −0.18, *CI* = −0.53–0.16, *p* = 0.304).

In summary, while music seemed to be irrelevant for participants with high working memory and high affective flexibility, it showed a relevant role for those with high working memory and low affective flexibility in distraction, which was carried out more successfully with music.

### 3.5. Reappraisal Mode with vs. without Music

First, we tried to identify the answers that fit into our predefined category: empathy. Empathy concerns the attempt to understand the other person’s feelings and emotions and comprehend nonjudgmentally the negative experiences of another self [45]. Some authors refer to two aspects of empathy: cognitive and affective. Cognitive empathy refers to the ability of a person to perceive things from the other’s point of view. Affective empathy concerns the human ability to vicariously share an emotion [46].

From the remaining answers, four categories emerged (see Figure 4): positive side/learning, relativization, detachment, and justification. In the positive side/learning category, participants tried to extract the positive aspects of the emotional situation and focus on what they learned from that. When they relativized, participants tried to introduce new elements that mitigate the importance of that situation and look at the situation in a non-absolute way. In detachment, participants tried to distance themselves from the emotional situation, seeing it from an outside point of view. Finally, in justification, participants tried to find a justification for the emotional situation.

The features that participants focused on in each reappraisal mode can be consulted in Appendix D.

In the R-W condition, participants dominantly reappraised the situation in empathic ways. There seemed to be an increase in cognitive empathy. In contrast, participants in the R-Wo condition focused mainly on trying to see the positive aspects and what they had learned to reappraise the situation.

When participants used empathy to reappraise the situation—dominant reappraisal mode under music—(*n* = 8), six of them (75%) stated that, in their perspective, the levels of anger would be lower if they remembered the situation again. When they focused on the positive aspects of the things they learned—the dominant reappraisal under no music (*n* = 6), only two of them (33.3%) said that the feelings of anger would not be the same. Thus, the reappraisal mode associated with music seemed to have been perceived as having more long-lasting effects than that associated with the absence of music.

As for the other question concerning the long-term efficacy of reappraisal (*Do you think you will be able to deal better with the emotional situation in the future?)*, only one participant (who tried to relativize the situation) responded in a negative way. Relativization was not dominant with music nor without music.

## 4. Discussion

This study had two main objectives. First, we aimed to test the facilitating effect of listening to music on the implementation of two specific ER strategies, distraction, and reappraisal, and if this effect is moderated by musical sophistication and/or executive functioning skills. Second, we aimed to determine whether music listening influences the way in which reappraisal is made, specifically whether music listening promotes empathic reappraisal modes.

### 4.1. Facilitating Effect: Does Music Make it Easier?

We started by testing the effects of listening to music, without considering musical sophistication or executive functioning as covariables. In line with Baltazar et al. [14], we observed no evidence that music facilitates the effectiveness of regulation strategies as measured by anger reports (objective measures of regulation). However, subjective measures (perceived success in emotional regulation) showed that participants who tried reappraisal with music reported fewer difficulties than those who did not listen to music.

There are several potential explanations for this apparent contradiction between the two measures of emotional regulation (objective vs. subjective). One relates to problems in objective measures of regulation and includes the possibility that the instrument used to measure state-anger (STAXI-2) failed to capture anger changes properly. For example, the instrument contained an item (“I feel bored”) that attempted to target the way participants felt at that moment due to remembering the annoying event. However, it is not impossible that participants reported their level of boredom resulting from participating in the experiment, instead of referring to the evoked emotional situation. Another explanation relates to possible frailties of subjective reports. The Social Cognitive Theory [22] postulates that music listening is guided by beliefs about the effects of music on emotional regulation. These beliefs can be created by experience and vicarious learning [9]. Therefore, if our participants used music to reappraise their emotional situation prior to the experiment, they might have gone through the session believing that music would help in ER, even if it did not. Future research could take into account participants’ beliefs related to music and emotional regulation. It could also consider using another self-report anger measure.

### 4.2. The Moderating Role of Musical Sophistication

Consistent with the idea that music is more relevant to higher than to lower musical sophistication individuals, our first finding was that music effects on objective measures of regulation were restricted to higher musical sophistication participants. Listening to music made a difference in the ER of higher musical sophistication participants for both distraction and reappraisal, but the effect of music was opposite across strategies: while music benefitted the implementation of reappraisal, it harmed the implementation of distraction. How can we account for this detrimental effect of music during distraction? One possible explanation is that, besides the need to be interesting/relevant, a distractive strategy should be complex and/or unexpected [6]. It is possible that individuals with higher musical sophistication tend to become increasingly selective and require very high levels of complexity or surprise to achieve distraction. Another possibility is that musically sophisticated individuals are more susceptible to the emotions induced by music [47] and are, therefore, more susceptible to emotional overloads that make the distraction task harder. According to Patston and Tippett [48], musicians show decreased performance in cognitive tasks during music listening. Also, in line with this idea of a trade-off between cognitive processing and emotionality, it has been shown that increases in cognitive load during a distracting activity can be related to decreased activation of the emotional system, and vice versa [6]. Regarding the positive effects of music in the reappraisal condition—which were limited to high-musical sophistication participants, it should be noted that these may not be fully in line with available evidence coming from music-therapy studies. For instance, [49] found that Cognitive Behavioral Therapy (CBT)—a therapeutic approach where reappraisal has a prominent role—combined with music is effective at relieving anxiety in students. However, the authors did not compare CBT alone with CBT combined with music (we do not know which component was critical: CBT or its combination with music), nor did they measure the musical sophistication of participants (they could be all musically sophisticated, in line with our findings). As for the null effect of music on the emotional regulation of less musically sophisticated participants that we found, it may at odds with evidence that music therapy sessions can be helpful to induce distraction in the elderly during clinical exams or therapy [50], though, once again, no measures of musical sophistication were presented in this study, and we cannot rule out that participants were highly musical. In general, our findings highlight the importance of considering musical sophistication when it comes to approaching emotional regulation aided by music. Though according to our findings, musical sophistication may modulate the outcomes of emotional regulation interventions, it has not often been considered, at least as much as it has been in studies addressing cognitive improvement (e.g., [51]).

### 4.3. The Moderating Role of Executive Functions

Taking the best model containing executive-functioning-related factors, we observed that working memory and a specific domain of affective flexibility (NA.N, the ability to switch into a neutral mode after being exposed to negative stimuli) moderated the effects of music.

Starting with working memory, participants with lower working memory levels showed detrimental effects of music, while those with higher levels showed benefits. In lower working memory participants, music made distraction more difficult. One explanation may be that lower working memory participants have more difficulty in keeping a goal in mind [52] the goal of achieving distraction. Because music promotes emotions and mind-wandering [6,53] music may have driven participants’ attention back to the problem (emotional situation) they were trying to ignore. Conversely, high working memory participants may have been more able to pursue their goal of getting distracted and avoiding problematic memory.

Music was either irrelevant or beneficial for distracting participants with higher working memory abilities. These different effects (null vs. beneficial) depended on effective flexibility. When effective flexibility was high, music was irrelevant; when low, it was beneficial. Unlike the previous moderating effect of working memory, the moderating effect of affective flexibility is in line with our hypothesis. When both working memory and affective flexibility were high, music was irrelevant maybe because participants had all the conditions to distract themselves without needing extra tools like music. In contrast, when participants had greater difficulties in changing into a non affective rule (focusing on the neutral features of the stimulus, in the presence of a negative stimulus), they were able to better distract themselves in the presence of music. The reason why music compensates for the lack of this executive functioning skill is not completely clear. One possible explanation is that the difficulty to focus on the neutral features within the negative stimulus can be overcome by focusing on the neutral features of music. According to this, participants in the distraction with music condition would possibly be focusing on the more neutral aspects of music, like rhythm.

In sum, we found evidence against, but also in favor of our hypothesis. While low levels of working memory do not seem to make participants use music as a support tool, low levels of affective flexibility in a specific domain have the effect we predicted, provided that higher levels of working memory are available. These results should be investigated further with larger and more controlled samples regarding executive functioning.

### 4.4. Reappraisal Mode: Does Music Make It Different?

Regarding the second objective of this study, we observed differences between music-related conditions in the way participants reappraised the emotional situation. Most participants used empathy-based reinterpretations in the reappraisal with music conditions. It is well-known [21] that prosocial songs exert an influence on participants’ feelings and thoughts, leading to increased empathy. Interestingly, in our study, we found that the musical features that participants engaged with when using reappraisal were mostly the feelings and memories evoked by the song. Although all songs had lyrics, lyrics were not the dominant driver of empathy (Appendix D).

The reason why music per se, without words, could generate empathy is not clear. Empathy is a process and not an emotional state, and various components in music may lead to a greater sense of empathy, such as mimicry and cognitive processes [53]. Mimicry concerns the predisposition to mimic the perceived emotional states of others, increasing, in turn, the predisposition to empathize with them. When listening to music, we tend to mimic the emotional state expressed by the song [54]. Considering that this mimic-related process is automatic and does not require higher-order cognitive processes [54], it is possible that mimicked emotions occur. Another potential explanation is that music increases the efficacy of cognitive processes underlying empathy, like perspective taking and inferences about the other’s mental state or intentions. Through visual imagery, music can lead to the creation of scenarios that make the emotional situation more prone to reappraisal. This visual imagery can co-occur with episodic memory, allowing people to visualize a specific event in their lives. These specific events, evoked by music, are usually related to social relations and can serve a nostalgic function [55]. In such cases, music could lead to the remembering of past positive events with the person that is now involved in the anger situation, leading to a better understanding of the situation and the way things happened the way they did. Future research could investigate these hypotheses, contributing to shed more light on the relationship between music and empathy. It is known that music therapy allows anger management and the improvement of coping strategies in forensic psychiatric patients [56], which might be related to the link between music listening and empathy that our study highlighted.

Participants who reappraised the situation in an empathic way answered yes more frequently to the questions “*In the future, if you remember that emotional situation again, the levels of anger evoked by her would be the same?*”. Future research could test further these preliminary results by investigating if this effect does extend in time and the reappraisal of that situation worked upon integration of participants’ life narratives, or, on the contrary, if these feelings of empathy and understanding are only evoked when listening to music. Even if participants who reappraised the situation in a more emphatic way were those that listened to music, this does not mean that there is a correlation.

### 4.5. Other Limitations and Prospects

A major limitation of our study concerns anger measures. Our objective measures of regulation success were based on self-reports of anger. Self-report measures may be biased by social desirability or demand effects [57], so future studies should include psychophysiological measures of anger in their experimental procedure.

In addition, we did not control for the possibility that some participants did not have enough memory skills to evoke the feelings of anger, resulting in lower self-reported anger, and lower ER intensity. Even when participants had a personal situation to recall, the fact that they had to recall it twice could have led to a decrease in their emotional impact, resulting in contamination on ER intensity. The two latter limitations can be avoided in the future by using non-personal stimuli to induce emotions, such as emotional pictures or films.

Although we have explicitly asked participants to recall an event that made them feel angry, those events could have elicited other emotions that were not covered by the STAXI-2, leaving some emotional experiences unnoticed. So, future research could include self-reported measures that can evaluate a more diverse spectrum of emotions.

Concerning sample size, we recruited enough participants to capture a medium effect (see methods, a priori power analysis). Though this is a reasonable approach to power, a larger sample would increase the sensitivity of the design to small effects, which may have gone unnoticed in the present study. It is also worth noting that our sample consisted mostly of young female participants. Older people tend to be more efficient in regulating their emotions [58]. Also, the effectiveness of emotional regulatory strategies is less consistent in women due to sex hormone fluctuations [59]. Therefore, diversifying sample characteristics also stands as a challenge for the future.

### 4.6. Overview

The present study contributed to clarifying the influence of music on emotional regulation. Our findings agree with the concept of musical affordance [60]. According to it, the power of music comes from the dynamic interactions between music and the listener, and making effective use of music in emotion regulation requires perceptual, affective, and sensitive skills. So, maybe ER strategies are like a cognitive way of dealing with emotional events, whereas music can be used as a tactic, provided that there are cognitive resources and/or relations with music. In line with this, our findings showed that music was a useful tactic only for those who are more capable to engage with music in a flexible and effective way [17]. In the same way, music was a relevant tool for those who have trouble switching to a non affective rule in the presence of a negative stimulus but have the working memory skills to use music to mitigate this.

Moreover, our study highlighted the empathic way participants found to reappraising the emotional situation in the presence of music. So, maybe music does not necessarily reduce the emotional activation caused by anger, but it provides context to find a more meaningful interpretation for the emotional event and promotes the subjective experience of success in regulating emotions. As we observed, music seems to enhance the feeling that, in the future, the problem will have a lower effective resonance, especially when the new meaning assigned makes us understand and forgive others. So, maybe the best use of music does not involve the containment of our emotional experience *per se*, but the creation of an ideal atmosphere that allows us to integrate this emotional experience into our life narrative.

## 5. Conclusions

We are very susceptible to music, and the power it exerts on us is undeniable. The way music is described by lay people and experts makes us believe that it is like an omnipotent entity, capable of cheering us up, calming us down, acting like a comforting friend [10], and giving us chills [61]. In the present study, we highlighted some potentialities of music in emotional regulation that remind us of the power of music but, not less important, we also highlighted some limitations. Listening to music seems to serve an important regulatory function, but this seems to depend on the listeners’ characteristics such as musical sophistication, high working memory, and low effective flexibility. Moreover, music seems to elicit a particular way of reappraisal based on empathy.

## Figures and Tables

**Figure 1 behavsci-12-00199-f001:**
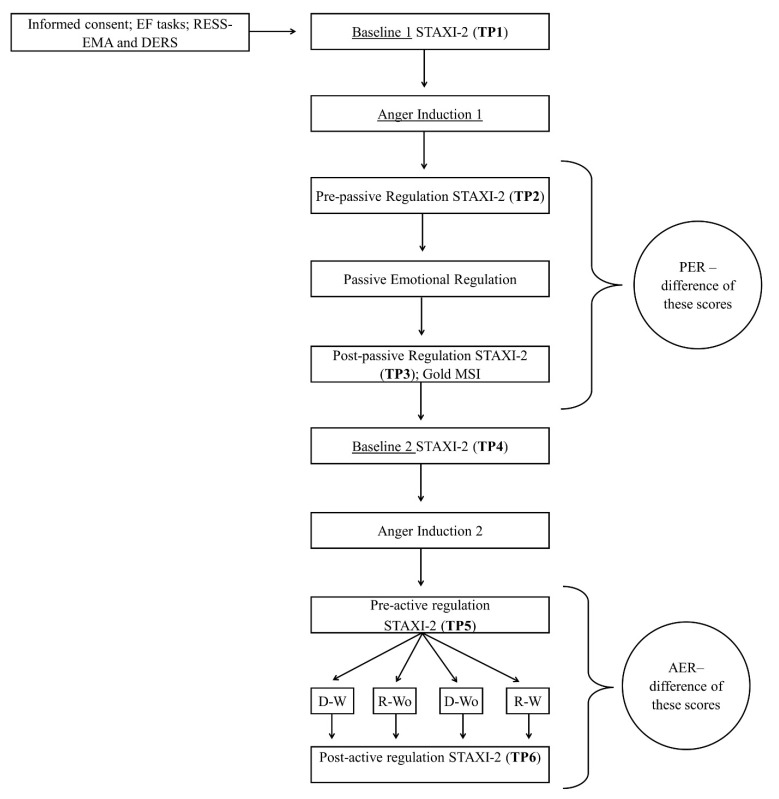
Experimental sequence (EF = Executive Functions; TP = Time Point; PER = Passive Emotional Regulation; AER = Active Emotional Regulation; D–W = distraction with music; D–Wo = distraction without music; R–W = reappraisal with music; R-Wo = reappraisal without music).

**Figure 2 behavsci-12-00199-f002:**
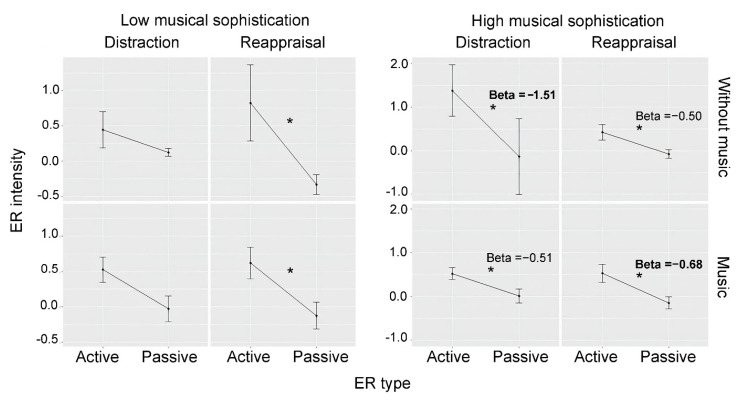
ER intensity according to ER type, music, and strategy (distraction vs. reappraisal) for Low vs. High-musical-sophistication participants. Asterisks indicate significant ER efficacy. Bold beta values point to increased effects of music. In low-musical-sophistication participants, music had no effects; in high-musical-sophistication participants, music benefitted reappraisal, but harmed distraction. Asterisks indicate significant ER efficacy. Bold beta values point to increased effects of music.

**Figure 3 behavsci-12-00199-f003:**
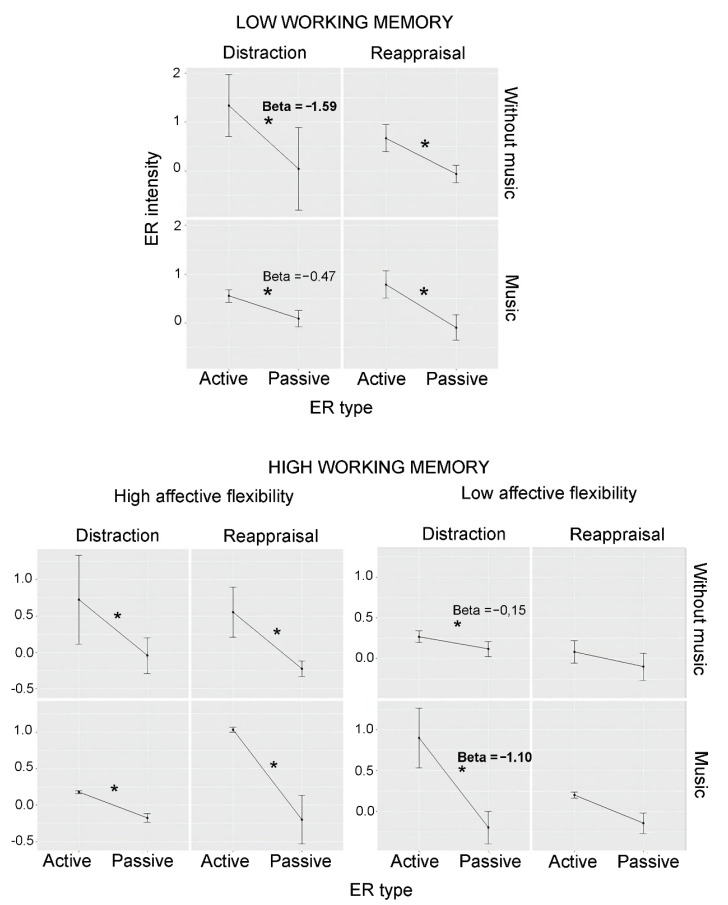
ER intensity according to ER type, music, and strategy (distraction vs. reappraisal) for low- vs. high-working-memory participants and, within the latter, for participants with high vs. low affective flexibility. In low-working-memory participants, distraction was decreased by music. In high-working-memory participants with low affective flexibility, distraction benefited from music. Asterisks indicate significant effects.

**Figure 4 behavsci-12-00199-f004:**
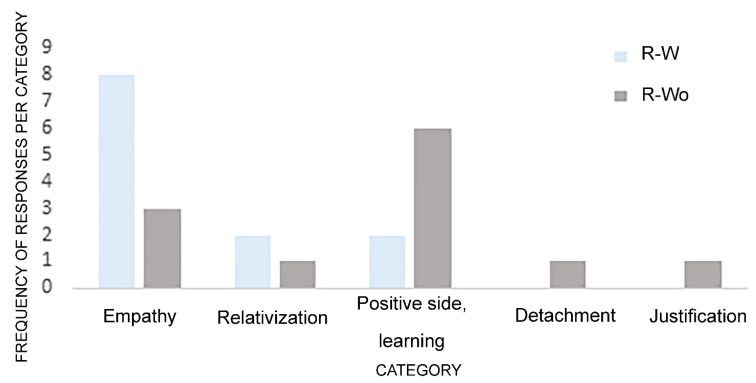
Number of answers for each reappraisal mode category (R-W = reappraisal with music; R-Wo = reappraisal without music).

**Table 1 behavsci-12-00199-t001:** Questionnaire on subjective perceptions of the regulation process.

Question [Target]	Type of Response
(1) Did you feel difficulties in the emotional regulation task? [How easy was the active emotional regulation task for participants]	Yes or no response
(2) How did you distract yourself/ reappraise the emotional situation? [Mode of distraction/ reappraisal] ^a^	Open response
(3) In the future, if you remembered that emotional situation again, would the levels of anger evoked by her be the same? [Projection of this emotional regulation into the future]	Yes or no response
(4) If not, would they be less or more intense?	Less or more response
(5) Do you think you will be able to deal better with the emotional situation in the future?	Yes or no response
(6) Which features of the song did you focus on? ^b^ − feelings evoked by the song − memories evoked by the song − rhythm and/or lyrics − other	Multiple choice (more than one allowed)

^a^ Question (2) was presented in either way, depending on the group (distraction vs. reappraisal). ^b^ Question (6) was presented only to participants in music groups.

**Table 2 behavsci-12-00199-t002:** Emotional regulation skills across experimental groups.

Instrument/Task	Mean (Standard Deviation)				One-Way ANOVA
	D-W	D-Wo	R-W	R-Wo	
*DERS*					
Total Score	105( 12.8)	105 (13.1)	105 (10.8)	102 (11.2)	*p* = 0.91, *η*^2^ = 0.01
Awareness	17.8 (3.21)	17.8 (1.78)	17.8 (2.5)	17.7 (2.56)	*p* = 0.99, *η*^2^ = 0.001
Clarity	9.71 (1.79)	9.71 (1.35)	10.9 (1.80)	10.1 (2.02)	*p* = 0.35, *η*^2^ = 0.072
Strategies	19.2 (3.43)	19.8 (2.04)	18.9 (2.14)	18.3 (2.58)	*p* = 0.57, *η*^2^ = 0.044
Impulse	12.9 (4.46)	13.2 (4.46)	12.6 (2.90)	12.6 (3.48)	*p* = 0.97, *η*^2^ = 0.005
Nonacceptance	20.3 (1.75)	19.9 (2.57)	19.8 (2.09)	19.4 (2.47)	*p* = 0.84, *η*^2^ = 0.019
Goals	25.4 (7.92)	25.1 (7.47)	24.9 (4.96)	24.3 (6.63)	*p* = 0.98, *η*^2^ = 0.004
*RESS EMA*					
Rumination	6.7 (1.36)	7.86 (1.49)	7.75 (1.48)	7.83 (1.62)	*p* = 0.26, *η*^2^ = 0.086
Relaxation	7.4 (1.35)	6.43 (1.43)	7 (1.83)	7.08 (1.26)	*p* = 0.47, *η*^2^ = 0.055
Reappraisal	7.1 (1.18)	7.36 (1.51)	7.33 (1.75)	8.42 (0.95)	*p* = 0.13, *η*^2^ = 0.119
Engagement	7.3 (1.67)	7.36 (1.85)	6.67 (1.80)	6.92 (1.66)	*p* = 0.76, *η*^2^ = 0.026
Distraction	7.2 (1.84)	7.57 (1.83)	7.5 (1.19)	8 (1.22)	*p* = 0.71, *η*^2^ = 0.030
Suppression	4.6 (1.80)	5.14 (1.62)	6.08 (1.80)	6.17 (1.77)	*p* = 0.14, *η*^2^ = 0.115

*Note.* DERS = Difficulties in Emotional Regulation Scale; RESS EMA = regulating emotion systems in everyday life inventory.

**Table 3 behavsci-12-00199-t003:** STAXI-2 ratings (mean and standard deviation) across all time points and resulting values of anger induction and emotional regulation intensity (TP = Time Point; asterisks indicate significant differences from zero).

	TP1	TP2	TP3	TP4	TP5	TP6
Mean (SD)	1.11 (0.15)	1.81 (0.69)	1.88 (0.67)	1.16 (0.21)	1.79 (0.76)	1.21 (0.28)
Anger Induction	(first, TP2-1) 0.70(0.64) *			(second, TP5-4 ) 0.62 (0.67) *		
Emotional Regulation Intensity			(passive, TP4-3) −0.07 (0.47)		(active, TP6-5) 0.58 (0.59) *	

**Table 4 behavsci-12-00199-t004:** Effects of ER type, music, and strategies on emotional regulation intensity.

	Emotional Regulation Intensity		
*Predictors*	*Estimates*	*CI*	*p*
(Intercept)	0.72	0.40–1.05	<0.001
ER type [passive]	−0.68	−1.11–−0.25	0.002
strategy [R]	−0.20	−0.64–0.24	0.369
music [Y]	−0.20	−0.63–0.22	0.355
ER type [passive] * strategy [R]	0.02	−0.57–0.60	0.957
ER type [passive] * music [Y]	0.14	−0.42–0.71	0.617
strategy [R] * music [Y]	0.26	−0.34–0.86	0.391
(ER type [passive] * strategy [R]) * music [Y]	−0.20	−0.99–0.59	0.622
SD (Intercept)	0.18		
SD (Observations)	0.70		
Random Effects			
σ^2^	0.24		
τ_00 ID_	0.03		
ICC	0.12		
N _ID_	48		
Observations	96		
Marginal R2/Conditional R2	0.286/0.370		

*Note.* Marginal R^2^ = variance explained by fixed factors; Conditional R^2^ = variance explained by fixed and random factors; bold = *p* < 0.05.

## Data Availability

The database used in current study is available at osf link https://osf.io/db4qh/?view_only=690239289ca34c1dbe770338a5b031ea (Appendix A).

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
