# Peer review of "The “Ifs” and “Hows” of the Role of Music on the Implementation of Emotional Regulation Strategies"

_behavsci, 2022, doi:10.3390/bs12060199_

Round 1

Reviewer 1 Report

This study investigated the influence of music listening on emotion regulation. It attempted to reveal how the influence of music on emotion regulation differ for people with different levels of music sophistication and executive functioning skill. Each participant went through two sessions of emotion regulation, one passive and one active, after anger is evoked using an autobiographical memory task. The level of anger was measured at various points in the course of experiment. For the active regulation task, participants were divided into four groups by the regulation strategy (destruction or reappraisal) and the presence of music (with or without music). The results showed that listing to music does not always facilitate emotion regulation, and the impacts depend on the listeners’ music sophistication and executive functioning skill.

This manuscript is well written and easy to follow despite the complexity of the experiment. The experiment is well thought out to test the authors’ hypotheses, and each variable was thoroughly and properly measured. The analyses were also thorough. However, it is better to have more information in some parts of the method section, and I have some concerns about how executive functioning were analysed.

  1. Please explain more about participant-experimenter interactions in the zoom experiment. Did participant see the experimenter on the screen all the time? It is important to address because the awareness of being seen by others might affect emotional experiences participants had during the experiment. Also, did the experimenter view what participants typed for their emotional episodes? What did participants actually do to “elaborate” on the episode at the second emotional evoking session? Did they add something to what they already typed?
  2. How long was the time gap between the TP3 and TP4?
  3. The top left section of Figure 1 should say RESS-EMA and DERS rather than saying “questionnaire”. Since there were many other questionnaires not presented in this time period, it is confusing.
  4. More information is needed for the active regulation task. Please provide some examples of distracting activities without music. How long did the task take? Was the duration equal across all groups?
  5. I think it is important to emphasize whether the music used in this experiment was songs with lyrics (I’m assuming that they were, but am I correct?). Also, if the music was with words you may mention that in the discussion section 4.4.
  6. Were all the questions on Table 1 presented for all participants? Some questions seem to be irrelevant for participants in particular group, for example, question 2 is irrelevant to the distraction groups and so is question 6 for the without music groups.
  7. On the result section 3.4, I have some concerns. First of all, why inhibitory control was not included in the analysis? Second, please explain why you look at the three types of (you only looked into two of them in the end) executive functioning separately. Wouldn’t is better to make some sort of a composite score from the three measurements and use it as an executive control index for each participant? Unless you have different hypothesis for each type of executive functioning, I don’t see the point of looking into them separately. This part makes already complicated experiment/analysis much more complex. If you investigate them separately, some justifications or separate hypothesis are necessary to be included in the introduction. Also, please report correlation coefficient among the three measurements. The relevant parts in discussion should be rewritten in accordance with the change made to the introduction/analysis.

  1. Minor mistakes

All figures should be in a better resolution.

Table 2 should be adjusted so that it won’t have unnecessary line brakes.

There are possible mistakes somewhere in each of the following lines.

Missing a space – line 49, 59, 555

Unnecessary space – line 41, 570, 638

Small font for names – 540, 671

Unnecessary bracket – line 466

Author Response

R1. The “Ifs” and “Hows” of the Role of Music on the Implementation of Emotional Regulation Strategies

RESPONSE TO COMMENTS

Reviewer 1

This study investigated the influence of music listening on emotion regulation. It attempted to reveal how the influence of music on emotion regulation differ for people with different levels of music sophistication and executive functioning skill. Each participant went through two sessions of emotion regulation, one passive and one active, after anger is evoked using an autobiographical memory task. The level of anger was measured at various points in the course of experiment. For the active regulation task, participants were divided into four groups by the regulation strategy (destruction or reappraisal) and the presence of music (with or without music). The results showed that listing to music does not always facilitate emotion regulation, and the impacts depend on the listeners’ music sophistication and executive functioning skill.

This manuscript is well written and easy to follow despite the complexity of the experiment. The experiment is well thought out to test the authors’ hypotheses, and each variable was thoroughly and properly measured. The analyses were also thorough. However, it is better to have more information in some parts of the method section, and I have some concerns about how executive functioning were analysed.

  1. Please explain more about participant-experimenter interactions in the zoom experiment. Did participant see the experimenter on the screen all the time? It is important to address because the awareness of being seen by others might affect emotional experiences participants had during the experiment. Also, did the experimenter view what participants typed for their emotional episodes? What did participants actually do to “elaborate” on the episode at the second emotional evoking session? Did they add something to what they already typed?

R: We have now clarified these issues: participant and experimenter only saw each other when instructions were being provided; the experimenter did not see what participants typed (lns 254-256); the elaboration consisted in adding text to the previously typed material (lns 224-225).

  1. How long was the time gap between the TP3 and TP4?

R: We have now specified this at lns…: it was the time they took to fill in the questionnaire, which was about 5 minutes (lns 251-252).

  1. The top left section of Figure 1 should say RESS-EMA and DERS rather than saying “questionnaire”. Since there were many other questionnaires not presented in this time period, it is confusing.

R: Done.

  1. More information is needed for the active regulation task. Please provide some examples of distracting activities without music. How long did the task take? Was the duration equal across all groups?

R: We have now specified distraction activities (Appendix E). We also specified the duration of the regulation tasks (5 minutes in all groups) - both passive and active - which was important information that was missing (lns 230 and 242-243).

  1. I think it is important to emphasize whether the music used in this experiment was songs with lyrics (I’m assuming that they were, but am I correct?). Also, if the music was with words you may mention that in the discussion section 4.4.

R: Yes, that is right. We clarified that they used music with lyrics in the procedure section (lns 242) as well as in the discussion (lns 580-581).

  1. Were all the questions on Table 1 presented for all participants? Some questions seem to be irrelevant for participants in particular group, for example, question 2 is irrelevant to the distraction groups and so is question 6 for the without music groups.

R: You are completely right, thanks. We have now made changes to the table to clarify that these two questions did not apply to all groups

  1. On the result section 3.4, I have some concerns. First of all, why inhibitory control was not included in the analysis? Second, please explain why you look at the three types of (you only looked into two of them in the end) executive functioning separately. Wouldn’t is better to make some sort of a composite score from the three measurements and use it as an executive control index for each participant?

 Unless you have different hypothesis for each type of executive functioning, I don’t see the point of looking into them separately. This part makes already complicated experiment/analysis much more complex. If you investigate them separately, some justifications or separate hypothesis are necessary to be included in the introduction. Also, please report correlation coefficient among the three measurements. The relevant parts in discussion should be rewritten in accordance with the change made to the introduction/analysis.

R: These are indeed important points, which we left unclear in the previous version.

As we have now clarified in the text, there were two reasons why we did not approach executive functioning with a composite measure. First, the literature suggests that, despite the common designations, executive functions may be independent from one another (e.g., inhibition dissociates from shifting), and this tends to occur mostly in adults. We added some references on that (lns 97-99). Second, our results did not show any evidence of association between the tasks: we ran both correlations between all tasks and an exploratory factor analysis supporting this, which we have now added to supplementary materials (S2). Therefore, we kept the tasks separate (see lns 313-317).

Regarding the absence of the Stroop task (inhibitory control) in the analysis, it was due to the method we used when choosing a model to analyse the moderating role of executive functions. As we have now clarified (lns 318-320), our method consisted of 1) implementing different models with different specifications regarding executive functioning variables (one function per model and, then, combinations of two functions) and 2) comparing all models to choose the one with the best fit. As we have now documented in S3 (Supplementary materials), the model with the best fit was the current one - including working memory and negative non-affective switch costs (one dimension of affective flexibility) as covariables.

Minor mistakes

All figures should be in a better resolution.

R: Done.

Table 2 should be adjusted so that it won’t have unnecessary line brakes.

R: The table has been formatted to fit into a single page.

There are possible mistakes somewhere in each of the following lines.

  • Missing a space – line 49, 59, 555
  • Unnecessary space – line 41, 570, 638
  • Small font for names – 540, 671
  • Unnecessary bracket – line 466

R: Unfortunately, the lines you mention do not seem to match the lines we used to number our submitted version. However, we have carefully revised the text for all details.

-------

We are grateful to the Reviewers for their insightful comments.

Reviewer 2 Report

The sample should be bigger and many references that the authors use are quite old; the authors should replace them with more recent ones.

I consider it to be a very interesting article that provides great information to society.

Author Response

R1. The “Ifs” and “Hows” of the Role of Music on the Implementation of Emotional Regulation Strategies

RESPONSE TO COMMENTS

Reviewer 2

The sample should be bigger

R: We have now incorporated the topic of sample size in the discussion (lns 620-622). Although we followed an a priori power analysis to determine sample size, we did this for medium effect sizes, and future studies could certainly increase sample size to capture smaller effects (in case we are dealing with these).

and many references that the authors use are quite old; the authors should replace them with more recent ones.

R: We have replaced several references by more recent ones (please see reference list, highlights in blue).

I consider it to be a very interesting article that provides great information to society.

-------

We are grateful to the Reviewers for their insightful comments.

Reviewer 3 Report

This paper examines the effect of music listening on emotion regulation mixed 2x2x2 factorial design study. The participants were divided into four groups based on regulation strategy (distraction vs. reappraisal) and music (with vs. without) and with two main objectives. 1) Evaluate the moderating effects exerted by musical sophistication and executive functioning on emotion regulation; 2) Evaluate the level of reappraisal in music vs non-music listeners.

Honestly, although I compliment the authors for designing such an original study, I really think that the analyses are too complex for presenting them in one publication. It is really hard to follow the aims of the paper, to understand the study design as well as the statistical analyses conducted to accomplish the two study aims. I think that this study would benefit if the results were split into more papers. Moreover, I suggest discussing also the potential clinical implications of the findings by integrating the discussion with findings from music therapy studies.

Author Response

R1. The “Ifs” and “Hows” of the Role of Music on the Implementation of Emotional Regulation Strategies

RESPONSE TO COMMENTS

Reviewer 3

This paper examines the effect of music listening on emotion regulation mixed 2x2x2 factorial design study. The participants were divided into four groups based on regulation strategy (distraction vs. reappraisal) and music (with vs. without) and with two main objectives. 1) Evaluate the moderating effects exerted by musical sophistication and executive functioning on emotion regulation; 2) Evaluate the level of reappraisal in music vs non-music listeners.

  1. Honestly, although I compliment the authors for designing such an original study, I really think that the analyses are too complex for presenting them in one publication. It is really hard to follow the aims of the paper, to understand the study design as well as the statistical analyses conducted to accomplish the two study aims. I think that this study would benefit if the results were split into more papers.

R:  We have simplified, whenever possible (e.g., lns 375-383), the structure of the results, such that they are easier to follow.

  1. Moreover, I suggest discussing also the potential clinical implications of the findings by integrating the discussion with findings from music therapy studies.

Thanks for your suggestion. We integrated the discussion with studies that highlighted the role of music in a therapeutic set (see lns 530-544 and 596-598).

-------

We are grateful to the Reviewers for their insightful comments.

Round 2

Reviewer 3 Report

I have no further comments